# COVID-19 Severity and Neonatal BCG Vaccination among Young Population in Taiwan

**DOI:** 10.3390/ijerph18084303

**Published:** 2021-04-18

**Authors:** Wei-Ju Su, Chia-Hsuin Chang, Jiun-Ling Wang, Shu-Fong Chen, Chin-Hui Yang

**Affiliations:** 1Centers for Disease Control, Ministry of Health and Welfare, Taipei 100, Taiwan; wei-ju@cdc.gov.tw (W.-J.S.); sfchen@cdc.gov.tw (S.-F.C.); 2Department of Internal Medicine, National Taiwan University Hospital, Taipei 100, Taiwan; chiahsuin123@yahoo.com.tw; 3Department of Internal Medicine, National Cheng Kung University Hospital, Tainan 701, Taiwan; jiunlingwang@Gmail.com

**Keywords:** BCG, Bacillus Calmette–Guérin, COVID-19, SARS-CoV-2, coronavirus disease 2019, vaccination, severity

## Abstract

Background: Data have not been reported to explore the relation between COVID-19 severity and BCG vaccination status at the individual patient level. Methods: Taiwan has a nationwide neonatal BCG vaccination program that was launched in 1965. The Taiwan Centers for Disease Control established a web-based National Immunization Information System (NISS) in 2003 and included all citizens’ BCG vaccination records in NISS for those born after 1985. We identified COVID-19 Taiwanese patients born after 1985 between 21 January and 19 March 2021. Study participants were further classified into ages 4–24 years (birth year 1996–2016) and 25–33 years (birth year 1986–1995). We described their clinical syndrome defined by the World Health Organization and examined the relation between the COVID-19 severity and BCG vaccination status. Results: In the 4–24 age group, among 138 BCG vaccinated individuals, 80.4% were asymptomatic or had mild disease, while 17.4% had moderate disease, 1.5% had severe disease, and 0.7% had acute respiratory distress syndrome but none of them died. In contrast, all 6 BCG unvaccinated individuals in this age group experienced mild illness. In the 25–33 age group, moderate disease occurred in 14.2% and severe disease occurred in 0.9% of the 106 patients without neonatal BCG vaccination records, as compared to 19.2% had moderate disease and none had severe or critical disease of the 78 patients with neonatal BCG vaccination records. Conclusions: Our finding indicated that BCG immunization might not relate to COVID-19 severity in the young population.

## 1. Introduction

The coronavirus disease 2019 (COVID-19) pandemic caused by severe acute respiratory syndrome coronavirus 2 (SARC-CoV-2) has led to an imminent need for an effective vaccination to constrain viral spread and reduce global disease morbidity and mortality. Bacillus Calmette–Guerin (BCG), a live attenuated vaccine against tuberculosis, was demonstrated to have non-specific immunomodulatory effects, by training the innate immune system to generate memory, which aided the host in fighting a wide range of viral infections in subsequent years [1,2]. BCG was reported to reduce viremia, respiratory tract infections, and neonatal mortality [1,2], but it was not tested effectively in a recent, large randomized trial [3]. In a small randomized double-blind placebo-controlled trial of hospitalized elderly patients, an un-prespecified interim analysis showed that 25.0% in the BCG group had new infections during the 12-month period of follow-up compared to 42.3% in the placebo group (*p* = 0.039), with most of the benefit coming from the reduction in respiratory tract infections of probably viral origin [4]. Whether BCG vaccination could have a beneficial effect against COVID-19 also under intense debate. Several ecological studies did suggest the potential protective effect of the universal BCG vaccination program on reducing COVID-19 morbidity and mortality [5,6,7,8,9,10,11,12,13]; some also implied that different BCG strains might vary in their protective ability [14]. In contrast, the inverse relation between COVID-19 disease burden and BCG vaccination was not observed by others [15,16]. However, these results are based on epidemiological data and have inherent biases [17,18]. Of note, all these studies analyzed aggregated data at the population level without directly comparing individuals that did receive BCG vaccinations to those that did not receive BCG vaccinations. A past study found that the trained immunity status induced by BCG immunization was maintained for at least one year [19], and BCG vaccination-induced protection against tuberculosis may last for approximately 20 years and waned thereafter [20]. Supposed BCG vaccination can produce cross-reactivity against COVID-19; theoretically, observational studies should evaluate the risk of COVID-19 during the first three decades after BCG immunization. Here, we described the nationwide neonatal BCG vaccination program in Taiwan and examined the relation between neonatal BCG vaccination and the COVID-19 clinical severity among young population.

## 2. Materials and Methods

Taiwan’s BCG vaccination program was launched in the 1950s. Originally, the program covered only school children with negative tuberculin skin test (TST) results, but it was extended to all neonates in 1965. At first, neonates were vaccinated with the Pasteur-1173 P2 strain, but in 1979, vaccinations shifted to the Tokyo-172 strain [21]. In 1965–1997, booster BCG vaccinations were given to 12-year-old adolescents with negative TST results. The BCG vaccine coverage rate increased to 87% in 1975 [22], and it has remained at 95.7–98.8% since 1996.

For the present study, records of BCG vaccinations in patients with COVID-19 were obtained from the web-based National Immunization Information System (NIIS), established by the Taiwan Centers for Disease Control (Taiwan CDC) in 2003. Electronic records for all publicly funded vaccines were compiled for all citizens of Taiwan born as early as the 1980s. However, an individual BCG vaccination record was unavailable in NIIS for those born in 1965–1985, and some were incomplete for those born in 1986–1995. The Taiwan CDC surveyed the NIIS database and found that the completeness of public-funded childhood immunization electronic records was above 90% for individuals born after 1996.

This study identified all COVID-19 cases in Taiwan born after 1985 through the Taiwan National Notifiable Disease Surveillance System between 21 January and 19 March 2021.

Confirmed cases fulfilled the criteria of notification for COVID-19 in Taiwan and their throat swab and sputum specimens were tested positive for SARC-CoV-2 RNA identification by real-time reverse transcriptase polymerase chain reaction (RT-PCR) test according to the Chinese protocol from Taiwan CDC. (2019-nCoV Virus Nucleic Acid Test. Available online: https://www.cdc.gov.tw/File/Get/BIHQoIEBjlFZ5tsjfij2Gg (accessed on 18 April 2020). Study participants were further classified into ages 4–24 years (birth year 1996–2016) and 25–33 years (birth year 1986–1995). We described participants’ COVID-19 severity according to the clinical syndrome defined by the World Health Organization [23,24]. (i.e., mild disease: patients with uncomplicated upper respiratory tract viral infection with non-specific symptoms. Moderate disease: pneumonia but no signs of severe pneumonia and no need for supplemental oxygen. Severe disease: fever or suspected respiratory infection, plus one of the following: respiratory rate >30 breaths/min; severe respiratory distress; or SpO_2_ ≤ 93% on room air. Critical disease: respiratory failure or septic shock).

Participants were classified as positive BCG vaccination status if their BCG vaccination records were available in the NIIS.

Fisher’s exact test was used to examine whether there were significant differences of COVID-19 disease severity (asymptomatic, mild disease, moderate/severe/critical disease) between participants with BCG vaccination records versus those without the records.

## 3. Results

A total of 625 Taiwanese patients with COVID-19 were identified during the study period. Their median age was 33.5 years (range 3–95 years), and 311 (49.8%) were female. Among them, 554 (88.6%) reported traveling outside Taiwan within 14 days of disease onset.

Of the total 625 confirmed COVID-19 cases during the study period, 328 (52.5%) were born after 1985.

Among them, 270 (82.3%) were asymptomatic or only had mild disease. Participants aged 4–24 years (birth year 1996–2016; *n* = 144) appeared to have a similar proportion of moderate, severe, and critical disease (17.4%, 1.5%, and 0.7%) as compared with those aged 25–33 years (birth year 1986–1995; *n* = 184) (16.3%, 0.5%, and 0%, respectively). Table 1 shows the relation between COVID-19 severity and BCG vaccination status.

Overall, there was no substantial difference in the COVID-19 severity between participants with BCG vaccination records versus those without the records (*p* = 0.25). Among 138 individuals aged 4–24 years that received neonatal BCG vaccinations, 111 (80.4%) were asymptomatic or had mild disease, while 24 (17.4%) had pneumonia (moderate disease), 2 (1.5%) had severe pneumonia (severe disease), and 1 (0.7%) had critical diseases, and none of them died. In contrast, all six individuals in this age group who did not receive neonatal BCG vaccinations experienced asymptomatic infection or mild disease. For participants aged 25–33 years, pneumonia occurred in 15 (14.2%) and severe diseases occurred in 1 (0.9%) of the 106 patients without neonatal BCG vaccination records, as compared to 15(19.2%) had pneumonia and none (0%) had severe or critical disease among the 78 patients with neonatal BCG vaccination records (*p* = 0.75). However, these data should be interpreted cautiously due to the fact that electronic vaccination records for the 25–33 age group might be incomplete. The sex distribution was shown in Appendix A.

## 4. Discussion

Our study showed that for COVID-19 patients aged 4–33 years who received the Tokyo-172 BCG vaccinations at birth, 19.0% had moderate or severe pneumonia, 0.5% acquired intensive care unit admission, and none died. According to the recent reports, the hospitalization, ICU admission, and case–fatality rate in the US was 14.3–20.8%, 2.0–4.2%, 0.1–0.2% for patients aged 20–44 years, and 5.7–20%, 0.58–2.0%, 0% for those aged <18 years [25], respectively. These data suggested that the COVID-19 morbidity and mortality was similar in the young population in Taiwan where the universal neonatal BCG vaccination program has been implemented for decades and that in the US, which has never had such a program.

In the earlier months of the COVID-19 pandemic, there were tremendous geographical variations in the pace and the extent of its outbreak. The number of COVID-19 cases and deaths increased dramatically in North America and Western Europe than in Asia, Africa, and South America. Accordingly, there is a hypothesis that the substantial differences may be related to the universal BCG vaccination policies of these countries. Several ecological studies analyzed worldwide data before June 2020 and found that even after controlling for important country level factors such as temperature, total population size, population density, proportion of those aged 60 years or above, gross domestic product per capita, healthcare access and quality measures, COVID-19 testing numbers, and stringency level of the nationwide epidemic control policy, the BCG vaccination coverage rate and the total BCG vaccination implementation years among these countries are still significantly inversely correlated with the incidence and mortality of COVID-19 [5,6,7,8,9,10,11,12,13].

Indeed, same relation was found even when restricting the analyses to western European countries with more similar socioeconomic conditions [5]. However, with the expansion of the epidemic, the number of deaths due to COVID-19 has increased rapidly in countries in South America and South Africa, which all have high BCG vaccination coverage. Arlehamn SCL et al. conducted subsequent analysis of updated data and found that the protective association of the BCG vaccination was attenuated [26], suggesting the observed differences are likely to be due to the various stages of the epidemic spread in these countries. Meanwhile, one study also found that the BCG vaccination coverage was also strongly correlated with countries’ other infectious control polices against COVD-19 [27], implying the observed country-level association may be further confounded. It is very likely that an early, decisive governmental act, for example the strictness of lockdown policy to eliminate the epidemics, may also partially explain the limiting community spread of COVID-19 in those countries. There is strong evidence for the early success of the elimination approach in New Zealand, Taiwan, Hong King, and South Korea. Up to now, western Pacific countries still have a lower mortality rate from COVID-19 [28,29].

In contrast to ecological studies comparing different countries, studies compared the COVID-19 infection rate and mortality rate in citizens who were born before the cessation of mass BCG immunization with those who were born after the discontinuation of the program in the same country and found no dissimilarity between the two groups [30,31,32].

Hamiel U et al. [30] analyzed data from Israel and found no difference in the SARS-CoV-2-positive rate between individuals born during the period of mandatory BCG vaccinations and those born outside that period. De Chaisemartin C et al. assessed the effect of BCG vaccination on COVID-19-related outcome by comparing the birth cohorts born just before and just after 1975 when Sweden discontinued its newborn’s vaccination program [31]. They found the odds ratios (95% C.I.) for COVID-19 cases and COVID-19-related hospitalizations were 1.0005 (0.8130–1.1881) and 1.2046 (0.7532–1.6560), respectively. In an article discussing the relation between BCG coverage and national COVID-19-related outcome [32], Lerm M reported that in East Germany with rigorous BCG nationwide vaccination policy for newborns until 1990, the number of COVID-19 cases among those aged <31 years was not higher than that of the older age group.

In another article [33], Bluhm R and Pinkovskiy M conducted geographic regression discontinuity analysis and found that the large differences in COVID-19 infection rates between the former East and West Germany were not attributable to BCG vaccination rates, but to the West’s copious commuter flow patterns. However, in these studies, the mean age of the BCG vaccinated cohort is apparently higher than that of the unvaccinated cohort. In fact, the clinical manifestations and mortality of COVID-19 varies substantially by age. The increased comorbidities of the elderly population also contribute to the high mortality rate of COVID-19.

Two recently published studies examined the association between BCG vaccination and COVID-19 incidence at the individual patient level [34,35]. An analysis of data from 1966 Italian doctors found that the infection rate of COVID-19 in physicians who had received BCG vaccination was 2.17%, similar to that of 1.66% in physicians who had never received BCG vaccination [34]. In contrast, another large seroprevalence study that analyzed data from 6201 healthcare workers in a multisite Los Angeles healthcare organization observed inconsistent findings [35]. The seroprevalence rate of anti-SARS-CoV-2 IgG was 2.7% among those who reported to have BCG vaccination history by questionnaire, marginally significantly lower than that of 3.8% of those who did not have BCG vaccination history (*p* = 0.044). However, it was unclear whether only subjects with milder disease severity were enrolled in the study, which may lead to sampling bias, and the history of BCG vaccination was obtained by questionnaire with undetermined accuracy.

To our knowledge, the present study analyzing real-world data, is the first research mainly targeting the young population for the relation between neonatal BCG vaccination and COVID-19 severity. The BCG vaccination records came from the national vaccination registry and hence the accuracy should be higher than the questionnaire. We included all cases with a full spectrum of severity to reduce the possibility of sampling bias. The severity of COVID-19 was categorized in detail according to the WHO criteria. However, these data should be interpreted cautiously, because the numbers of cases in the vaccinated and unvaccinated groups were small, some vaccination records in childhood might be incomplete, and host genetic factors and SARS-CoV-2 virulence factors might have had a substantial influence on the COVID19 outcome.

Our study contributes to the existing literature about the null effect of BCG vaccination against COVID-19. On this occasion, while most countries all over the world are actively implementing a large-scale COVID-19 vaccination plan, we would like to remind that optimal infection control policies and social ecological system [36,37,38], including the enforcement of containment measures of border control, quarantine policy, testing strategy of risk groups, intensive contact tracing, discouraging mass gathering, and the wide use of personal protection equipment, are still crucial in eliminating this disastrous pandemic.

## 5. Conclusions

In conclusion, our study, based on the individual patient data, did not find strong evidence that neonatal BCG vaccination has a substantial protective effect in reducing COVID-19 morbidity and mortality in the young age group in Taiwan.

## Figures and Tables

**Table 1 ijerph-18-04303-t001:** Clinical syndromes associated with COVID-19 stratified by participant’s Tokyo-172 BCG vaccination status among age 4–24 years and 25–33 years between 21 January and 19 March 2021 in Taiwan.

Birth Year (Age) *	without Neonatal BCG Vaccination Records						with Neonatal BCG Vaccination Records					
Severity	Asymptomatic	Mild	Moderate	Severe	Critical	Total	Asymptomatic	Mild	Moderate	Severe	Critical	Total
1996–2016 (4–24)	0	6 (100)	0	0	0	6	15 (10.9)	96 (69.6)	24 (17.4)	2 (1.5)	1 (0.7)	138
1986–1995 (25–33)	7 (6.6)	83 (78.3)	15 (14.2)	1 (0.9)	0	106	6 (7.7)	57 (73.1)	15 (19.2)	0	0	78
Total	7 (6.3)	89 (79.5)	15 (13.4)	1 (0.9)	0	112	21 (9.7)	153 (70.8)	39 (18.1)	2 (0.9)	1 (0.5)	216

* National Immunization Information System (NIIS) records were complete for those born after 1996 (age 24 years or younger). NIIS records might be incomplete for individuals born in 1986–1995 (age 25–33 years).

## Data Availability

The data presented in this study are contained within the article or contact the corresponding authors for detailed data.

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
