# Peer review of "COVID-19 Severity and Neonatal BCG Vaccination among Young Population in Taiwan"

_ijerph, 2021, doi:10.3390/ijerph18084303_

Round 1

Reviewer 1 Report

In this study, Su et.al analyzed the percentages of various COVID-19 syndromes in populations vaccinated with or without BCG. BCG vaccine has been shown to boost immunity, mostly non-specifically,  against infection and also cancer. Therefore, a rationale is behind exploring the link between BCG vaccination and COVID-19 incidence/disease severity, like several other studies did similarly. This study was appropriately designed, clearly presented, and well written. Together with several other studies, mixed results are shown regarding the link between BCG vaccination and COVID-19 incidence/disease symptoms, One possible reason is the datasets and the method used for comparisons. These have been discussed extensively by the authors. 

I have two major comment:

1-What is the rationale for classifying the participants into groups 4-24 years and 25-33 years? In the 4 to the 24-year-old and non-BCG vaccinated group, there were only 8 participants, comparing to 133 participants in the 4 to 24-year-old and BCG-vaccinated group. The numbers are rather unmatched. 

2-The participants' disease severity is described according to WHO. To help readers understand more easily, the symptoms for the classifications should be provided in detail in the methods section. 

Author Response

1-What is the rationale for classifying the participants into groups 4-24 years and 25-33 years? In the 4 to the 24-year-old and non-BCG vaccinated group, there were only 8 participants, comparing to 133 participants in the 4 to 24-year-old and BCG-vaccinated group. The numbers are rather unmatched. 

Ans: The rationale for classifying the participants into group 4-24 and 25-33 is due to the immunization record. In Taiwan,National Immunization Information System (NIIS) records were complete for those born after 1996 (age 24 years or younger). And NIIS records might be incomplete for individuals born in 1986-1995 (age 25-33 years).  We have add these two sentence in the footnote of Table. 

For the low case number , we have rechecked and extended our case series for another 3 months. The new result can be seen in Table. However, the case numbers without BCG in age 4-24 is still low.

2-The participants' disease severity is described according to WHO. To help readers understand more easily, the symptoms for the classifications should be provided in detail in the methods section. 

Ans: We have added the following sentences in the text(line 86-91). 

(i.e., Mild disease: Patients uncomplicated upper respiratory tract viral infection with non-specific symptoms. Moderate disease: pneumonia but no signs of severe pneumonia and no need for supplemental oxygen. Severe disease: fever or suspected respiratory infection, plus one of the following: respiratory rate > 30 breaths/min; severe respiratory distress; or SpO2≤ 93% on room air. Critical disease: respiratory failure or septic shock) 

Reviewer 2 Report

Authors used Taiwanese BCG data to demonstrate un-relatedness between neonatal vaccination and severity of COVID-19 in two age groups. They claim that younger cohort age 4-24 does not have merit of BCG vaccination. However, the sizes of cohorts are 133 for vaccinated vs only 8 for unvaccinated. These disparity and small size are concerning to draw any conclusions. Furthermore, sexes of these cohorts are not shown. Therefore, I advise to employ more samples in the study for a successful publication.

Author Response

Authors used Taiwanese BCG data to demonstrate un-relatedness between neonatal vaccination and severity of COVID-19 in two age groups. They claim that younger cohort age 4-24 does not have merit of BCG vaccination. However, the sizes of cohorts are 133 for vaccinated vs only 8 for unvaccinated. These disparity and small size are concerning to draw any conclusions. Furthermore, sexes of these cohorts are not shown. Therefore, I advise to employ more samples in the study for a successful publication.

Ans: We have increase our sized of cohort after including 3 months of cases after 2021. And about sex distribution, we have added another supplement table. 

Reviewer 3 Report

The analyses and discussions presented in the manuscript seem to support what others have found as per the references cited by the authors.

A couple of things I noticed. The authors said that they used Chi-square test (lines 89 - 92). I did not find the test result.

Second thing I noticed that the BCG-vaccinated participants in the age-cohort of 25-33 years (birth year 1986-1995; N = 142) appeared to be more protected from COVID-19 than the BCG-vaccinated participants in the age-cohort of 4 - 24 years. Why? Or it is not significant? Or the booster BCG shot received (see lines 66 - 67) by the participants in the age-cohort of 25-33 years might have been responsible? 

Author Response

The analyses and discussions presented in the manuscript seem to support what others have found as per the references cited by the authors.

1.A couple of things I noticed. The authors said that they used Chi-square test (lines 89 - 92). I did not find the test result.

Ans: We have added the Chi-square test in the test(LINE 121-4). "For participants aged 25-33 years, pneumonia occurred in 15 (14.2%) and severe diseases occurred in 1 (0.9 %) of the 106 patients without neonatal BCG vaccination records, as compared to 15(19.2 %) had pneumonia and none (0%) had severe or critical disease among the 78 patients with neonatal BCG vaccination records (p = 0.71)."

2.Second thing I noticed that the BCG-vaccinated participants in the age-cohort of 25-33 years (birth year 1986-1995; N = 142) appeared to be more protected from COVID-19 than the BCG-vaccinated participants in the age-cohort of 4 - 24 years. Why? Or it is not significant? Or the booster BCG shot received (see lines 66 - 67) by the participants in the age-cohort of 25-33 years might have been responsible? 

Ans: Due to the low case numbers, the difference may be not significant. And we don't think this is related to boost shot.

Round 2

Reviewer 2 Report

The cohort of 6 samples in 4-24 without BCG vaccination does not suffice to support authors' conclusion. Also, comparisons with US cohorts for disease severity does not make sense due to so many confounding factors such as differences in strain, host genetic and treatments.

Author Response

The BCG vaccination coverage rate in Taiwan is over 98%. So the case numbers without BCG is low. The limitation of this comparison is described in the discussion part.

"These data should be interpreted cautiously, because the numbers of cases in the vaccinated and unvaccinated groups were small, some vaccination records in childhood might be incomplete, and host genetic factors and SARS-CoV-2 virulence factors might have had a substantial influence on the COVID19 outcome."

 And due to the low case numbers , we have change our Chi-Square test to Fisher exact test

This manuscript is a resubmission of an earlier submission. The following is a list of the peer review reports and author responses from that submission.